# Infrared Small and Moving Target Detection on Account of the Minimization of Non-Convex Spatial-Temporal Tensor Low-Rank Approximation under the Complex Background

**Kun Wang** [1,2], **Defu Jiang** [1,*], **Lijun Yun** [2] **and Xiaoyang Liu** [3]

1   College of Computer and Information, Hohai University, Nanjing 211000, China
2   School of Information Science and Technology, Yunnan Normal University, Kunming 650500, China
3   Department of Computer Science and Engineering, Chongqing University of Technology, Chongqing 400054, China
*   Correspondence: surfer_jiangdf0801@163.com

**Featured Application: Authors are encouraged to provide a concise description of the specific application or a potential application of the work. This section is not mandatory.**

**Abstract:** Infrared point-target detection is one of the key technologies in infrared guidance systems. Due to the long observation distance, the point target is often submerged in the background clutter and large noise in the process of atmospheric transmission and scattering, and the signal-to-noise ratio is low. On the other hand, the target in the image appears in the form of fuzzy points, so that the target has no obvious features and texture information. Therefore, scholars have proposed many object detection methods for dimming infrared images, which has become a hot research topic on account of the flow-rank model based on the image patch. However, the result has a high false alarm rate because the most low-rank models based on the image patch do not consider the spatial-temporal characteristics of the infrared sequences. Therefore, we introduce 3D total variation (3D-TV) to regularize the foreground on account of the non-convex rank approximation minimization method, so as to consider the spatial-temporal continuity of the target and effectively suppress the interference caused by dynamic background and target movement on the foreground extraction. Finally, this paper proposes the minimization of the non-convex spatial-temporal tensor low-rank approximation algorithm (MNSTLA) by studying the related algorithms of the point infrared target detection, and the experimental results show strong robustness and a low false alarm rate for the proposed method compared with other advanced algorithms, such as NARM, RIPT, and WSNMSTIPT.

**Keywords:** complex background; infrared image; MNSTLA; point target detection

## 1. Introduction

The infrared detection system has the advantages of not being affected by light and, therefore, being capable of working at all times of the day [1]; not emitting electromagnetic waves and, therefore, being a system using a non-automatic detection method [2]; and having a strong penetrability and, therefore, being capable of penetrating the covers of dust, clouds, and smoke so as to better identify false camouflage targets, making it an effective supplement or substitute for the traditional visible light detection system and the radar detection system [3]. Therefore, the infrared point and moving target detection on account of the infrared detection system has always been an important topic and hotspot of research.

The infrared images have a low rank feature due to the many repetitive elements in the background, and they have a sparse feature due to the few feature points of infrared points and moving targets [4,5]. In this case, the detection of infrared points and moving targets is transformed into a classification task on account of the good performance of

sparse representation in the classification task, which is what the method on of low-rank sparse is concerned about.

A sparse representation-based multispectral image target detection method was first proposed by the US Army Sensor Research Laboratory in 2014 [6]. He adopted the augmented Lagrange multiplier method to perform the optimization on account of the SR theory and the low-rank matrix [7] in 2015 based off the LRSR mode. This method can detect dim and point targets in a background with strong noise but does not have a good background suppression effect.

To overcome the limitations of conventional methods, Gao put forward an IPI (Infrared Patch-Image) model on account of the image segmentation by means of a sliding window, and the method can detect dim and point targets according to the targeted sparse feature of each patch image [8]. Considering the non-local autocorrelation structure for the background, the assumptions of the infrared patch image (IPI) model are in excellent agreement with the true scenario, which rephrases:

$$D_P = B_P + T_P + N_P \tag{1}$$

where $D_P$, $B_P$, $T_P$, and $N_P$ are patch-images corresponding to the original, background, target, and random noise images, which are shown separately. Furthermore, the features of low-rank for the background $B$, and the target $T$, which is sparsity.

Dai et al. of Nanjing University introduced a structural prior model into the detection process of infrared points and moving targets, namely WIPI (Weighted Infrared Patch-Image). This method can better preserve the infrared point and moving targets while suppressing the strong edges [9]. Dai proposed an RIPT model. Furthermore, in view of the detection of infrared points and moving targets with insufficient prior information and strong edges [10], the SNN is used to separate the real target from the background by combining the non-local and local spatial priors. In order to solve the problems that the observation values of strong edge information are insufficient and the implicit assumptions do not match, The NIPPS model put forward by Dai, which can detect the residual error in the target image and is used for singular values [11]. As the SNN is not a convex envelope of low-rank background, and in view of the fact that the traditional IPT method only uses spatial information, Sun proposed the WNRIPT model [12].

In order to adapt to different images and solve the problem of images with strong edges, Xiong Bin used adaptive weights and an augmented Lagrange multiplier method [13]. Wang put forward an IPI model-based detection method for infrared point and moving targets, which maintains the spatial correlation among images, constructs a patch image form, and uses the ADMM multiplier method to optimize the solution finding so as to deal with the non-smooth and non-uniform background by the TV-PCP method [14].

Wang used different multi-subspaces for the areas to reduce the interference in each area, combined the APG with the patch coordinate descent method, and used the SMSL method to improve the accuracy of heterogeneous background [15]. However, for the infrared images with a complex background, and especially for those that contain clutter signals, as the noise also has a sparse feature as the target, the false alarm rate will increase. For complex scenarios, Zhang et al. put forward a non-convex rank approximation minimization (NRAM) detection method for infrared points and moving targets, which introduces extra regular terms into the edges [16]. Although the NRAM method has achieved good results in single image frame detection, the false alarm rate of this method is still high in complex and changing scenarios because it does not consider spatial and temporal information.

The above methods only vectorize the infrared image into a matrix, but do not well consider the temporal information. Therefore, many methods on account of the tensor analysis are applied in the IRST system, such as multi-view clustering [17], subspace clustering [18], super-resolution image generation [19], and image video processing [20]. Tensor analysis not only considers the spatial information of image sequences but also the temporal information thereof.

First, to fully exploit the inter-frame correlation between infrared image sequences, considering the time consistency and local spatial smoothness between the consecutive frames of the target, we introduced the spatial-temporal tensor into the NRAM model. To obtain more precise background estimations in the detection of infrared points and moving targets, as there was considerable noise in the infrared scenario, the norm was introduced because, compared with the norm, the norm requires not only sparse columns but also sparse rows, which can better remove the strong edge non-target noise. In order to simplify the computational complexity, we introduced the Frobenius norm. Finally, we proposed a minimization of the non-convex spatial-temporal tensor low-rank approximation algorithm (MNSTLA). The main contributions of the MNSTLA model are:

(1) A non-convex spatial-temporal tensor low-rank approximation minimization method for the detection of infrared points and moving targets in the sequence scenarios was proposed. We introduced 3D-TV regularization into the NRAM model. The 3D-TV constraint on the background is helpful for keeping the image details and removing the noise, so it can achieve better detection performance under complex backgrounds.

(2) The norm is introduced into the detection of IR points and moving targets to better describe the target components. By combining structured sparsity terms, non-target components, especially those with strong edges, can be eliminated.

(3) The ADMM is used to efficiently reduce the computational complexity and solve the low-rank component recovery problem.

The paper is organized as follows: in Section 2, the work related to the MNSTLA method-based detection of infrared dim and point targets is briefly described; in Section 3, the proposed MNSTLA model; in Section 4, the extensive experiments carried out on various sequence scenarios are described to illustrate the efficiency of the MNSTLA model, and the results are evaluated subjectively and objectively; and in Section 5, we give the discussion and conclusion.

## 2. Related Work

In this section, we first briefly introduce how to construct an image sequence into a spatial-temporal patch tensor model of image tensors. Furthermore, we introduce the 3D-TV regularization model and the tensor kernel norm model, respectively, and model the foreground and background of the sequence image tensor considering both models.

### 2.1. Spatial-Temporal Patch Tensor Model

Generally speaking, given an image sequence $f_1, f_2, \ldots, f_p \in R^{m \times n}$ and a cube patch tensor $F \in R$, the frames can be obtained by stacking them in time order. The tensor of the IR point target image can be expressed as:

$$D_T = B_T + N_T + T_T \tag{2}$$

where $D_T, B_T, N_T, T_T \in R^{m \times n \times L}$ present the original patch-tensor, background tensor, target-tensor, and noise-tensor. According to the infrared imaging mechanism, the relative motion between the imaging sensor and the target is usually due to small changes at a long distance, such as an early warning system. Therefore, it is generally believed that the backgrounds of different frames change slowly in the whole sequence images, which means that there is a correlation between adjacent sequences [8,21]. For the reason that images containing infrared points and moving targets are considered to be of low rank, the constructed background tensor can also be considered a low-rank tensor. Compared with the matrix model, constructing a tensor model can not only mine the internal relations between data from more angles in the tensor domain but also further improve the capability of target detection by combining the spatial-temporal information.

### 2.2. Foreground Modeling on Account of 3D-TV Regularization

Total variation (TV) regularization is widely used to detect the sharp edges and corners of images, which can represent the desired spatial smoothness. In this study, we use 3D-TV to leverage spatio-temporal information. Assuming $N \in R^{m \times n \times t}$, we define the 3D-TV norm as:

$$||T||_{3D-TV} = \sum_{m,n,t} TV_{m,n,t}(T) = |T_{m+1,n,t} - T_{m,n,t}| + |T_{m,n+1,t} - T_{m,n,t}| + |T_{m,n,t+1} - T_{m,n,t}| \tag{3}$$

where $T_{m,n,t}$ represents the intensity of the pixels *(m, n, t)*; at the same time, the difference operator along the temporal direction shows that it considers the persistence of the foreground target in time.

We introduced the vector difference operators for the horizontal, vertical and time directions:

$$\begin{cases} V_h||T|| = vec(|T_{m+1,n,t} - T_{m,n,t}|) \\ V_v||T|| = vec(|T_{m,n+1,t} - T_{m,n,t}|) \\ V_t||T|| = vec(|T_{m,n,t+1} - T_{m,n,t}|) \end{cases} \tag{4}$$

Then, the Formula (3) can be rewritten as:

$$||T||_{3D-TV} = ||VT||_1 = ||V_h T||_1 + ||V_v T||_1 + ||V_t T||_1 \tag{5}$$

### 2.3. Background Modeling on Account of the Tensor Nuclear Norm

In the TRPCA model [22], the tensor nuclear norm is usually used instead of the rank function to constrain the background. However, the general tensor nuclear norm is used to matrix the tensor, and using the singular value of matrix to define the tensor nuclear norm will destroy the spatial structure of the video, and the degree of approximation to the rank function will be insufficient. On account of the t-product, Lu, et al. [23] an improved tensor nuclear norm is proposed:

$$||B||_{**} = \sum_{i=1}^{r} S(i, i, 1) \tag{6}$$

where $r = rank_t(B)$, $B = U * S * V$. and converted into the nuclear norm of the matrix:

$$||B||_{**} = \frac{1}{N}||bcric(B)||_* = \frac{1}{N}||\overline{B}||_* \tag{7}$$

From the Formulas (6) and (7), we obtain:

$$||B||_{**} = \frac{1}{n_3} \sum_{i=1}^{r} \sum_{j=1}^{n_3} \overline{S}(i, i, j) \tag{8}$$

where $bcric(B)$ represents the patch cyclic matrix of $B$, and $\overline{B}$ represents the patch diagonal matrix of $B$.

It can be seen from the Formula (6) that the improved tensor nuclear norm is directly defined by the singular value tensor $S$, and it can be seen from the patch cyclic matrix and patch diagonal matrix of the Formula (7) that the above tensor nuclear norm is defined on account of the front-side slicing (the third-dimension time). In addition, the improved tensor nuclear norm $||B||_{**}$ is a convex envelope of the average rank in the unit sphere of the tensor spectral norm, which has a better approximation to the rank function [23] on account of the above considerations; this paper uses the above tensor nuclear norm to perform low-rank constraining on the background, which strengthens the low rank of the background.

### 3. Methods

The spatial-temporal infrared patch-tensor model is described as:

$$f_D = f_B + f_T + f_N \tag{9}$$

where $f_D$, $f_B$, $f_T$, and $f_N$ represent the original, background, target, and noise images, respectively. As shown in Figure 1, each image frame is split into small image patches, and all the small image patches of consecutive $L$ frames are superimposed into the 3D patch-tensor. Therefore, the above formula can be rewritten into a tensor form as shown in Formula (2) in Section 2.2.

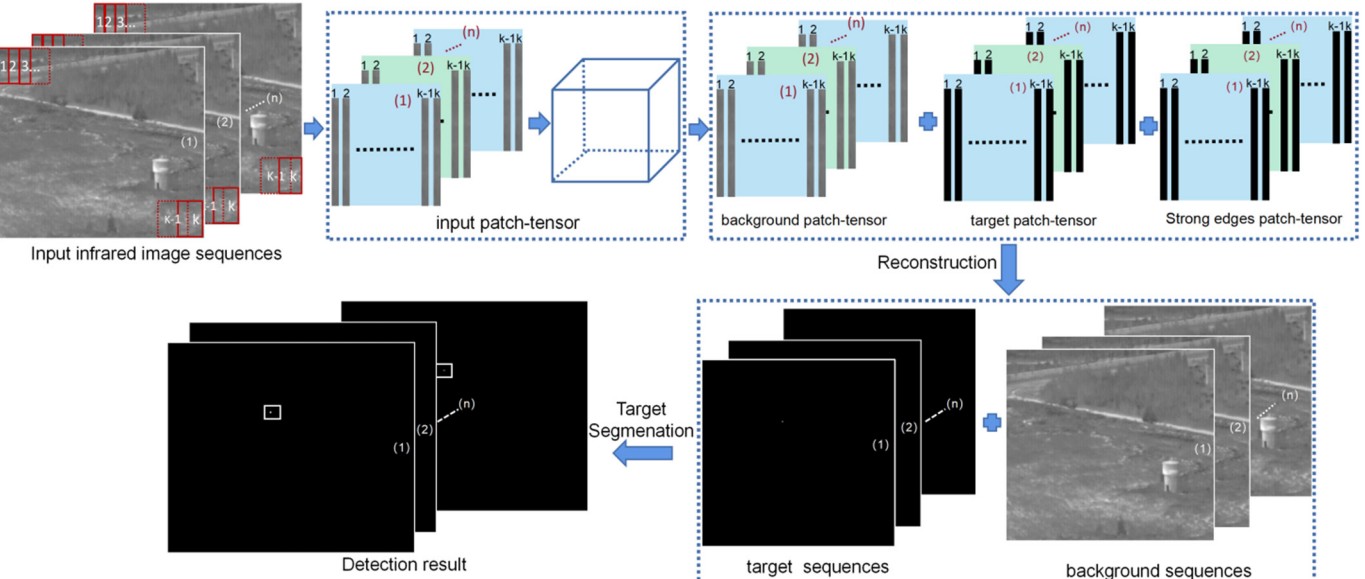

**Figure 1.** Flow Chart of the MNSTLA Method.

In the WNRIPT model, the problem of point target detection is expressed as:

$$B.T = \frac{min}{B.T} ||B||_{W_B,*} + \lambda ||W_T \odot T|| \tag{10}$$

where $\left|\left|B\right|\right|_{W_B,*} = \frac{1}{L} \sum_{i=1}^{r} \sum_{j=1}^{L} W_B(i,i,j)\overline{S}(i,i,j)$.

In order to further improve the performance and efficiency of point target detection, the 3D-TV regularization is introduced into the spatial-temporal tensor model, and its expression is:

$$B.T.N = \frac{arg\ min}{B.T.N} ||B||_{W_B,*} + \lambda_1 ||V(B)||_{3D-TV} + \lambda_2 ||T||_1 + \lambda_3 ||N||_F^2$$

$$s.t.\ F = B + T + N \tag{11}$$

where $k \times || * ||_{3D-TV}$ is the norm of 3D-TV, and $\lambda_1$, $\lambda_2$, and $\lambda_3$ represent the regularization parameters of the 3D-TV term, target component, and noise component.

As the Frobenius norm [24,25] has a good noise suppression effect, the Frobenius norm term is further introduced:

$$B.T.N = \frac{arg\ min}{B.T.N} ||B||_{W_B,*} + \lambda_1 ||V(B)||_1 + \lambda_2 ||T||_1 + \lambda_3 ||N||_F^2$$

$$s.t.\ F = B + T + N \tag{12}$$

In this model, the 3D-TV regularization term is introduced, which can fully capture the spatial-temporal information of infrared sequence images, so it is expected to achieve better performance.

### 3.1. Low Rank and Sparse Frame Model

Different values of singularities in the conventional convex kernel norm solve the imbalance penalty. Due to the equal treatment mechanism, if singular values are far from 1, the nuclear norm will have a considerable deviation. Each time the nuclear norm weight is determined, additional SVD will appear [26], which increases the running time of the method. Zhao proposed the $\gamma$ norma which is a new rank of non-convex function [27]. The $\gamma$ norm is unitarily invariant. The $\gamma$ norm is almost in agreement with the true rank ($\gamma = 0.002$), and the heuristic of the log-det performs poorly at minimal singular values [28], in particular when the value is close to 0; the $\gamma$ norm of the matrix $B$ is described as:

$$||B||_\gamma = \sum_i \frac{(1+\gamma)\sigma_i(B)}{\gamma + \sigma_i(B)} \tag{13}$$

For the reason the $l_0$ norm is NP-hard, the $l_1$ norm [29] assigns the same weight to each single element. Therefore, many other methods use the $l_1$-norm to characterize the sparsity of the target patch-image [30–32], and the target T with the $l_1$-norm is described as follows:

$$||T||_1 = \sum_{i,j} W_{ij}|T_{ij}| \tag{14}$$

where $W_{ij} = C / (|T_{ij}| + \varepsilon_T)$ is an element at position ($i$, $j$), $C$ is a compromise constant; moreover, $\varepsilon_T$ is a small positive number.

Infrared images also have a lot of strong edge noise, which makes many advanced methods [33–35] leave residual errors in the target image. The strong edge $E$ is linearly sparse relative to the whole image, and each line (i.e., line vector) is described by the vector $l_2$ norm, $w_i = \sqrt{\sum_j |E_{i,j}|^2}$, that is, the vector $w = [w_1, w, \dots, w_d]^T$, and then the whole matrix $E$ needs to be described by the norm. Therefore, the $l_1$ norm is used to describe $w$, that is, the $l_{2,1}$ norm of the strong edge $E$:

$$||E||_{2,1} = ||w||_1 = \sum_{i=1}^d \sqrt{\sum_{j=1}^n |E_{i,j}|^2} \tag{15}$$

According to the foregoing discussion, the patch-tensor model for the infrared image sequences is proposed on account of the minimization of the non-convex spatial-temporal tensor low-rank approximation algorithm (MNSTLA), that is, Formula (10) is redefined as:

$$B.T.E = \begin{array}{c} arg\ min \\ B.T.E \end{array} ||B||_{\gamma,*} + \lambda_1||L||_\gamma + \lambda_2||T||_1 + \lambda_3||E||_{2,1}$$

$$s.t.\ D = B + T + E \tag{16}$$

### 3.2. Solution Finding of MNSTLA Model

The optimization method based on the ADMM is used to work out Formula (16). Formula (16) can be rewritten as an augmented Lagrange function:

$$L(D, B, T, E, L, Z, Y, \mu)$$

$$= ||Z||_{\gamma,*} + \lambda_1||L||_\gamma + \lambda_2||T||_{w,1} + \langle Y_1, Z - B \rangle + \langle Y_2, L - V(B) \rangle + \langle Y_3, D - B - T - E \rangle$$
$$+ \frac{\mu}{2}\left(||Z - B||_F^2 + ||L - V(B)||_F^2 + ||D - B - T - E||_F^2\right) + \lambda_3||E||_{2,1}$$

$$s.t.\ D = B + T + E, Z = B, L = V(Z) \tag{17}$$

where $Y_*$, $\mu$ are an Lagrange multiplier and a positive penalty scalar, $\langle * \rangle$ represents the inner product, and $|| * ||_F$ is the norm for Frobenius.

The ADMM method is used to iteratively update the $Z$ and $L$ by the Formula (17), respectively:

$$Z^{k+1} = \underset{Z}{arg\ min}||Z||_{\gamma,*} + \frac{\mu^k}{2}||Z - B^k + \frac{Y_1{}^k}{\mu^k}||_F^2 \tag{18}$$

$$L^{k+1} = \underset{L}{arg\ min}||L||_\gamma + \frac{\mu^k}{2}||L - V(B^k) - \frac{Y_1{}^k}{\mu^k}||_F^2 \tag{19}$$

Find their solutions by t-SVD [20] operation and unit contraction operator, respectively:

$$Z^{k+1} = D_{W/\mu^k}(B^k - \frac{Y_1^k}{\mu^k}) \tag{20}$$

$$L^{k+1} = Th_{\lambda_1/\mu^k}\left(V(B^k) - \frac{Y_2{}^k}{\mu^k}\right) \tag{21}$$

where $D(*)$ represent the t-SVD operation and $Th(*)$ represent the unit contraction operator. Extract the term containing $B$ from the Formula (17):

$$B^{k+1} = \frac{\mu^k}{2}(||D - B - T^k - E^k + \frac{Y_1^k}{\mu^k}||_F^2 + ||Z^{k+1} - B + \frac{Y_2^k}{\mu^k}||_F^2 + ||L^{k+1} - V(B) + \frac{Y_3^k}{\mu^k}||_F^2) \tag{22}$$

The Formula (22) is equivalent to the following linear equations:

$$(2I + V(B))B^{k+1} = D - T^k - E^k + \frac{Y_1^k}{\mu^k} + Z^k + \frac{Y_2^k}{\mu^k} + V^T(V_B^k + \frac{Y_3^k}{\mu^k}) \tag{23}$$

The closed form of the Formula (23) can be obtained by 3D Fast Fourier Transform:

$$B^{k+1} = ifftn(\frac{fftn(D - T^k - E^k + \frac{Y_1^k}{\mu^k} + Z^k + \frac{Y_2^k}{\mu^k} + V^T(V_B^k + \frac{Y_3^k}{\mu^k}))}{2\mu^k I + \mu^k|fftn(V(B)|^2}) \tag{24}$$

where $fftn$ is the fast 3D Fourier transform and $ifftn$ is the inverse transform of the $fftn$. Variables $T$ and $E$ are corrected:

$$T^{k+1} = \underset{T}{arg\ min}\lambda_2||T||_{W,1} + \frac{\mu^k}{2}||D - B^{k+1} - T - E^k + \frac{Y_3^k}{\mu^k}||_F^2 \tag{25}$$

$$E^{k+1} = \underset{E}{arg\ min}\lambda_3||E||_{2,1} + \frac{\mu^k}{2}\left|\left|D - B^{k+1} - T^{k+1} - E\right|\right|_F^2 \tag{26}$$

By using the element-by-element shrinkage operation method in references [29,36], we obtain:

$$T^{k+1} = Th_{\lambda W/\mu^k}(D - B^{k+1} - E^k - \frac{Y_3^k}{\mu^k}) \tag{27}$$

$$E^{k+1} = \frac{\mu^k(D - B^{k+1} - T^{k+1} - \frac{Y^k}{\mu^k}) + Y_3^k}{\mu^k + 2\lambda_3} \tag{28}$$

### 3.3. The Processing of the MNSTLA

The steps of the MNSTLA model (Algorithm 1):

---

**Algorithm 1:** The Minimization of Non-Convex Spatial-Temporal Tensor Low-Rank Approximation Algorithm(MNSTLA)

---

**Input:** Input the $f_1, f_2, \ldots, f_p \in R^{m \times n}$, $\lambda_1, \lambda_2, \lambda_3$, $L$ and $tol = 10^{-7}$
**Initialize:** Original patchtensor $D \in R^{m \times n \times L}$, $B^0 = T^0 = E^0 = Y_1{}^k = Y_2{}^k = Y_3{}^k = 0, \mu = 1e - 2$
ADMM for solving the Equation (17)
　　　while
　　　　　(1) Fix the others and update  and $L$by (20) and (21) $Z^{k+1}, L^{k+1}$
　　　　　(2) Fix the others and update $B$ by (24) $B^{k+1}$
　　　　　(3) Fix the others and update $T$by (25) $T^{k+1}$
　　　　　(4) Fix the others and update $E$ by (26) $E^{k+1}$
　　　　　(5) Check the convergence conditions $\frac{||D - B^{k+1} - T^{k+1}||_F^2}{||D||_F^2} \leq tol$
　　　　　(6) Update $k = k + 1$.
　　　Output $B^{k+1}, T^{k+1}$

---

The flow chart of the MNSTLA model is shown in Figure 1.
The specific detection steps are as follows:

(1)　The original infrared image sequences $f_1, f_2, \ldots, f_p \in R^{m \times n}$ are sequentially arranged by $n_3$ adjacent frames and are converted into several patch-tensor tensors $D \in R^{m \times n \times L}$.
(2)　The original patch-tensor is decomposed into the target patch-tensor $T$, background patch-tensor $B$, and structural noise (strong edge) patch-tensor $E$ by using the method 1.
(3)　The target image $I_T$ and the background image $I_B$ are reconstructed by inverse operation.
(4)　In the last step, we segment the target using the adaptive threshold [8]:

$$t_{seg} = mean(C) + \lambda \times std(C) \tag{29}$$

where $mean(C)$ is the mean value of the reconstructed confidence map, $std(C)$ is the standard deviation, and $\lambda$ is a constant.

## 4. Experiment and Analysis of Experimental Results

Where $mean(C)$ is the mean value of the reconstructed confidence map, $std(C)$ is the standard deviation, and $\lambda$ is a constant.

*4.1. Data Set and Evaluation Indicators*

4.1.1. Test Data Set

In the experiment, the "A data set for infrared detection and tracking of dim-small aircraft targets underground/air background [37]" collected by Hui Bingwei et al. was used. The sensors used for data acquisition were refrigerated medium-wave infrared cameras with a resolution of $256 \times 256$ pixels.

There are 22 data scenarios in this dataset. The *22* image sequences of data 1–data 22 of this data set data are described and shown in Table 1:

**Table 1.** Detailed Description of 22 Real Scenarios.

| Data | No. Frame | Scenario Description |
|---|---|---|
| data1 | 399 | Close range, single target, sky background |
| data2 | 599 | Close range, two targets, sky background, cross flight |
| data3 | 100 | Close range, single target, air-ground interface background, the target enters the field of view again after leaving the field of view. |
| data4 | 399 | Close range, two targets, sky background, cross flight |
| data5 | 3000 | Long range, single target, ground background, long time |
| data6 | 399 | From near to far, single target, ground background |
| data7 | 399 | From near to far, single target, ground background |
| data8 | 399 | From far to near, single target, ground background |
| data9 | 399 | From near to far, single target, ground background |
| data10 | 401 | Target from near to far, single target, ground-air interface background |
| data11 | 745 | Target from far to near, single target, ground background |
| data12 | 1500 | Target from far to near, single target, target mid-course maneuver, ground background |
| data13 | 763 | Target from near to far, single target, dim target, ground background |
| data14 | 1462 | Target from near to far, single target, ground background, target interfered by ground vehicles |
| data15 | 751 | Single target, target maneuver, ground background |
| data16 | 499 | Target from far to near, single target, extended target, target maneuver, ground background |
| data17 | 500 | Target from near to far, single target, dim target, ground background |
| data18 | 500 | Target from far to near, single target, ground background |
| data19 | 1599 | Single target, target maneuver, ground background |
| data20 | 400 | Single target, target maneuver, air-ground background |
| data21 | 500 | Long range, single target, ground background |
| data22 | 500 | Target from far to near, single target, ground background |

As can be seen from the above table, data1–data 4 all have a sky background. As they have a single background and large targets as shown by Figure 2, they are not suitable for our set conditions and are not used.

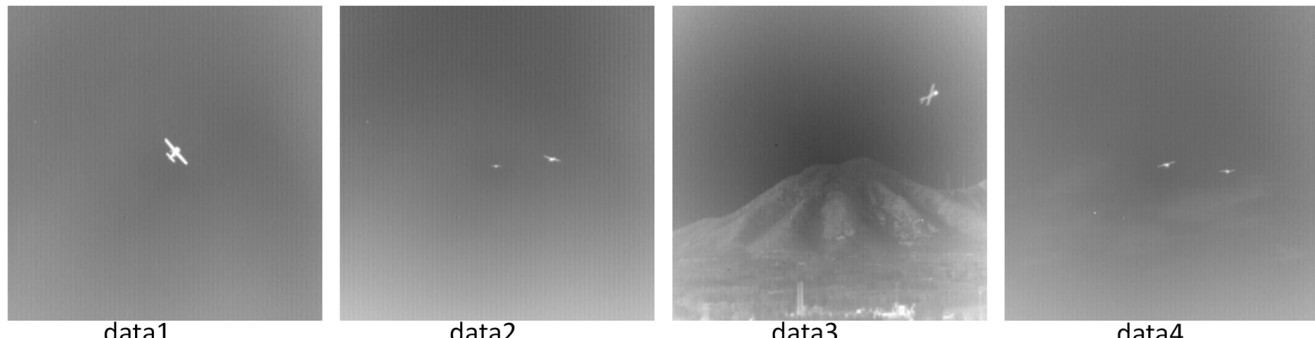

Figure 2. Data1–data 4 Sequence Images.

Six sequences of data 6, data10, data13, data14, data17 and data 22 were selected from data 5–data 22 as the sequence images of our experiment. As shown by Figure 3a–f, they are six representative images in the six sequences of the selected six data sets, namely, data 6, data 10, data 13, data 14, data 17, and data 22. The point-target is in the white boxes.

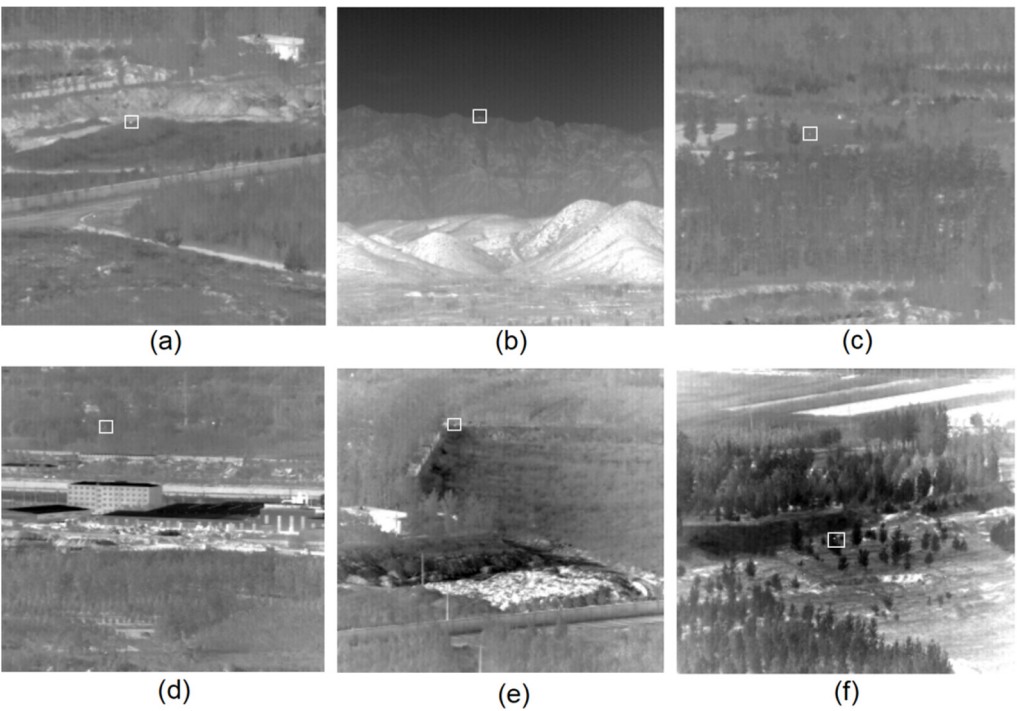

**Figure 3.** Six Infrared Image Sequences Selected.

4.1.2. Evaluation Indicators

The performance of dim object detection methods is generally evaluated using three criteria: background suppression, target enhancement, and detection accuracy.

(1)  Background suppression factor (*BSF*) [9]:

The *BSF* is defined as follows:

$$BSF = \frac{\delta_{out}}{\delta_{in}} \tag{30}$$

where $\delta_{out}$ and $\delta_{in}$ represent the local background standard deviation around the target of the output image and the original image.

(2)  Local contrast gain (LCG)

The *SCRG* represents the signal and noise ratios (*SCR*) before and after processing:

$$SCRG = \frac{SCR_{out}}{SCR_{in}} \tag{31}$$

In which the *SCR* uses the same expression as in reference [38]:

$$SCR = \frac{|\mu_t - \mu_b|}{\delta_b} \tag{32}$$

where $\mu_t$ , $\mu_b$ and $\delta_b$ represent the average gray values of the targets in the image.

In this paper, both *BSF* and *SCRG* need the determination of the background range around the target. Figure 4 shows the background around the target calculated in this paper, where d takes 20.

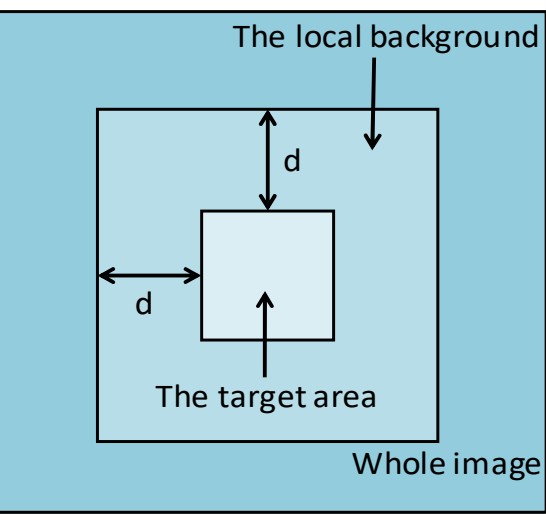

**Figure 4.** Local Background around the point targets in the Infrared Image.

For the reason the $\delta_b$ is close to zero in the Formula (32), it is difficult to evaluate the performance as the *SCR* approaches infinity. Therefore, we evaluate the performance of the target augmentation using *LCG*:

$$LCG = \frac{LC_{out}}{LC_{in}} \tag{33}$$

$$LC = \frac{|\mu_t - \mu_b|}{\mu_t + \mu_b} \tag{34}$$

where $LC_{out}$ and $LC_{in}$ represent the local contrast (*LC*) of the output image and the input image, the $\mu_t$ and $\mu_b$ the are consistent with those in the Formula (32).

(3)    Receiver operating characteristic curve (ROC)

In order to further compare the methods, the ROC curve is used to evaluate the methods which can be used to select the best category judgment model and abandon the sub-optimal model. When judging the category, the ROC curve can give a correct evaluation without being limited by cost or benefit.

All the samples, which is actually the target but is wrongly judged. It is defined as follows:

$$P_d = \frac{N_{true}}{N_{act}} \tag{35}$$

$$P_f = \frac{N_{false}}{N_{img}} \tag{36}$$

where $N_{true}$, $N_{act}$, $N_{false}$ and $N_{img}$ represent the number of really detected targets, the actual targets, the falsely detected targets and the frames, respectively.

### 4.2. Parameter Setting

We quote the values of $\mu$, $\gamma$, and *C* in reference [16], which are the penalty factor $\mu = c\sqrt{min(m,n)}$, where *c* = 3, $\gamma$ = 0.002, and *C* = 2.5, where *m* and *n* are the length and width of patch images, respectively. References [39–41] all made a detailed analysis of the frame number *L*, and we also take its value and the frame number *L* = 3. For details, please refer to these references.

In order to better verify the advancement of the MNSTLA method, we will compare it with seven advanced methods, including the Top-Hat method. Table 2 lists the parameter settings for these methods.

**Table 2.** The parameters for the 7 tested methods.

| Methods | Parameter Setting |
|---|---|
| Top-Hat | Structure size: $3 \times 3$, structure shape: square |
| PSTNN | Sliding step : $40$, $\lambda = 0.6/\sqrt{max(n_1, n_2) * n_3}$, patch size: $40 \times 40$, $\varepsilon = 1 \times 10^{-7}$ |
| IPI | Patch size : $50 \times 50$, sliding step : $10$, $\lambda = 1/\sqrt{min(m,n)}$, $\varepsilon = 10^{-7}$ |
| RIPT | Patch size : $30 \times 30$, $\lambda = L/\sqrt{min(m,n)}$, sliding step: $10$, $L = 0.7$, h = 1, $\varepsilon = 10^{-7}$ |
| WSNMSTIPT | Patch size : $30 \times 30$, sliding step : $30$ $L = 6$, p = 0.8, $\lambda = 1/\sqrt{max(n_1, n_2) * n_3}$ |
| NRAM | Patch size : $50 \times 50$, sliding step : $10$, $\lambda = 1/\sqrt{min(m,n)}$, $\mu 0 = 3\sqrt{min(m,n)}$, $\gamma = 0.002$, $C = \sqrt{min(m,n)}/2.5$, $\varepsilon = 10^{-7}$ |
| MNSTLA | Patch size : $50 \times 50$, sliding step : $10$, $\gamma = 0.002$, $\mu = c\sqrt{min(m,n)}$ where c = 3, $L = 3$. C = 2.5, $\varepsilon = 1 \times 10^{-7}$ |

### 4.3. Subjective Evaluation in Different Scenes

In this sub-section, we give the detection results of six infrared image sequences. The method proposed herein is compared with six related advanced methods, namely Top-Hat [41], IPI [9], PSTNN [22], IPT [23], WSNMSTIPT [24], and NRAM [16]. For the convenience of observing the results, the experimental results obtained and the three-dimensional grid diagrams generated by all the test methods in different scenarios are given intuitively in Figures 5–10.

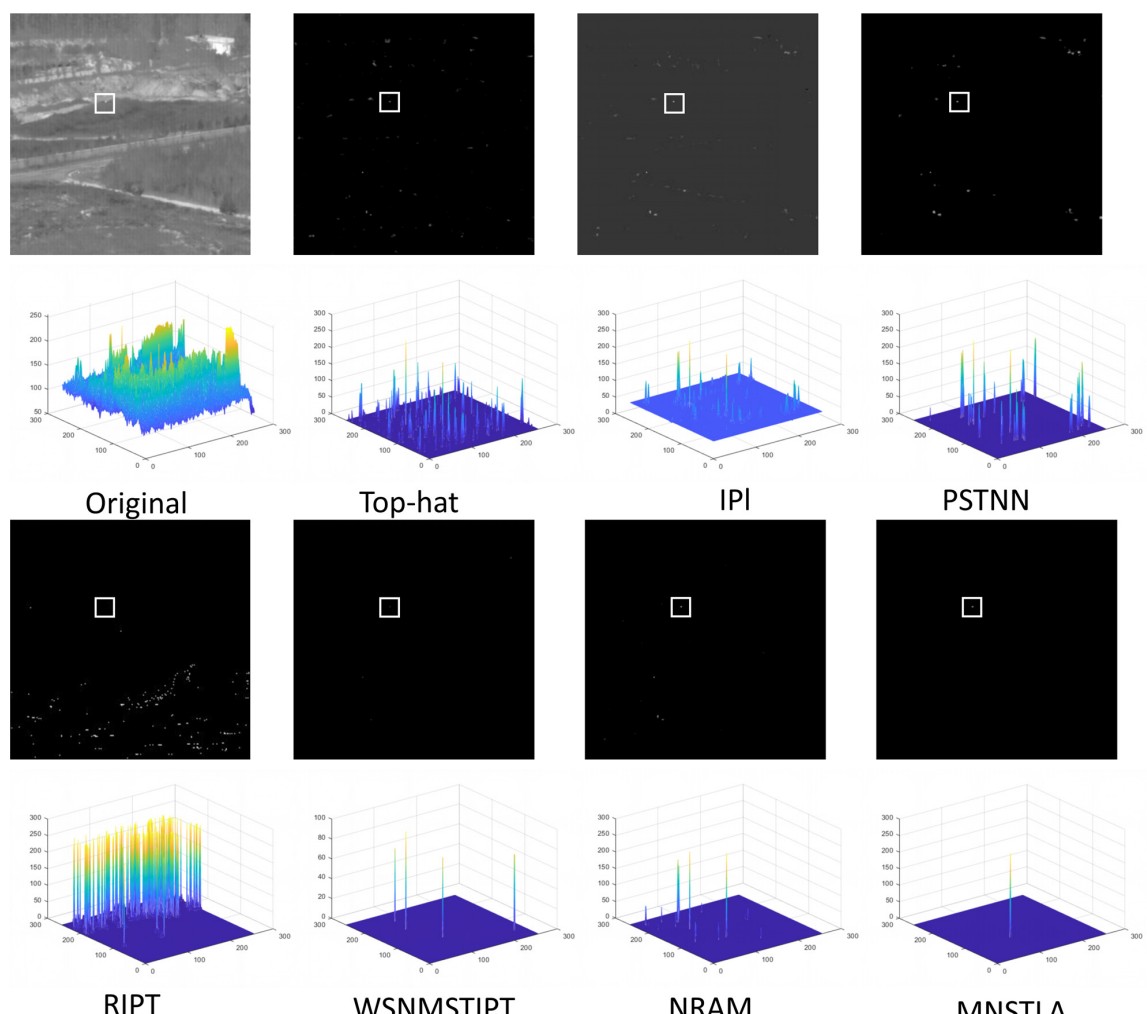

**Figure 5.** Infrared Sequence (a) Original image and Detection Results and the 3d grid diagrams.

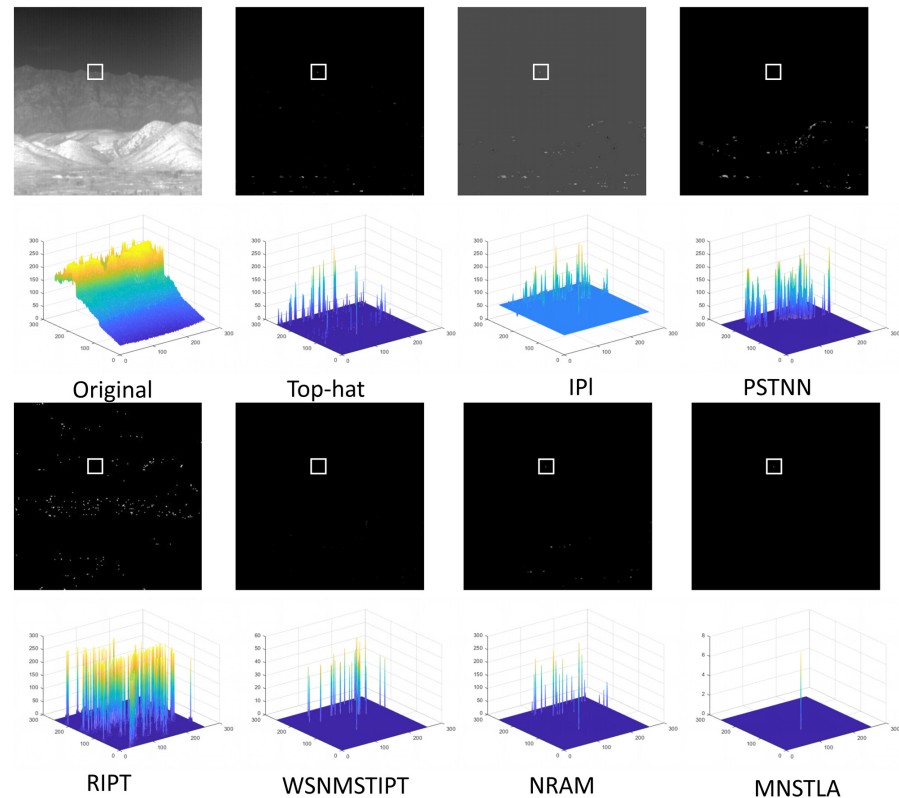

**Figure 6.** Infrared Sequence (b) Original image and Detection Results and the 3d grid diagrams.

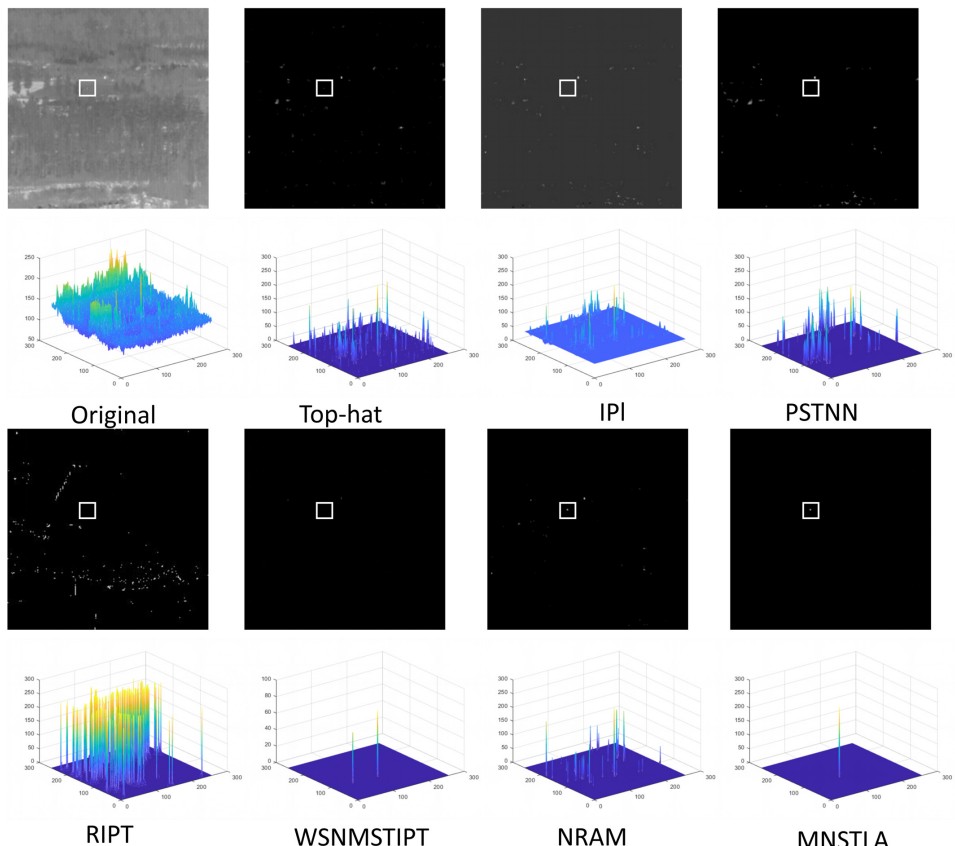

**Figure 7.** Infrared Sequence (c) Original image and Detection Results and the 3d grid diagrams.

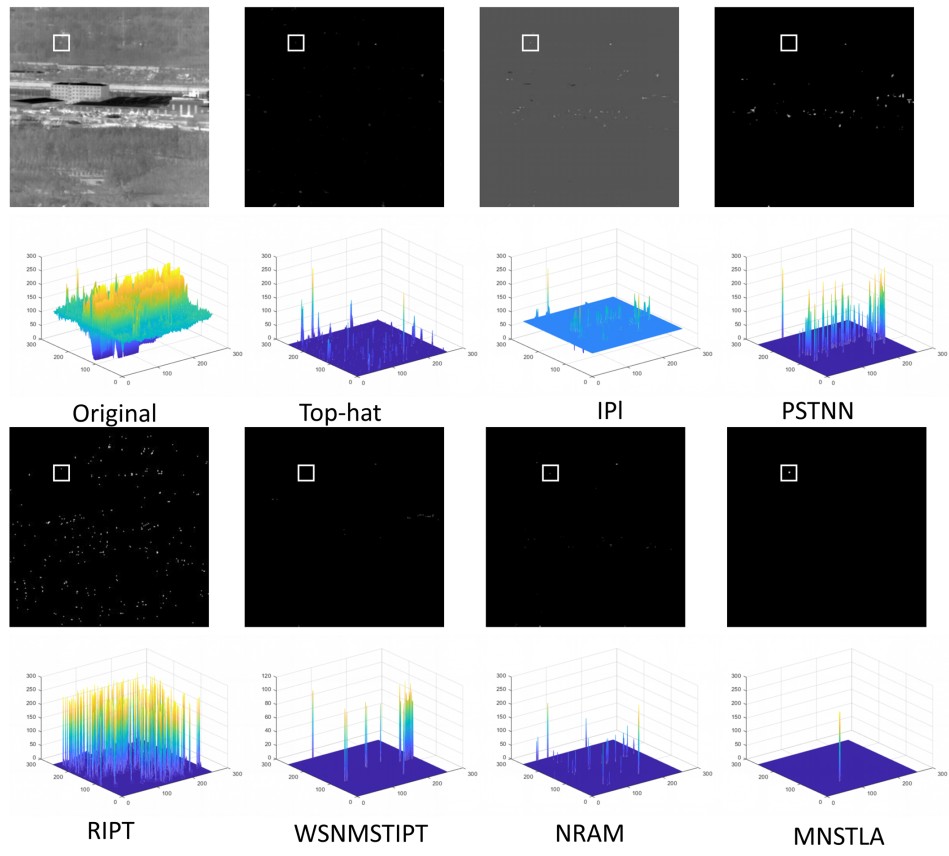

**Figure 8.** Infrared Sequence (d) Original image and Detection Results and the 3d grid diagrams.

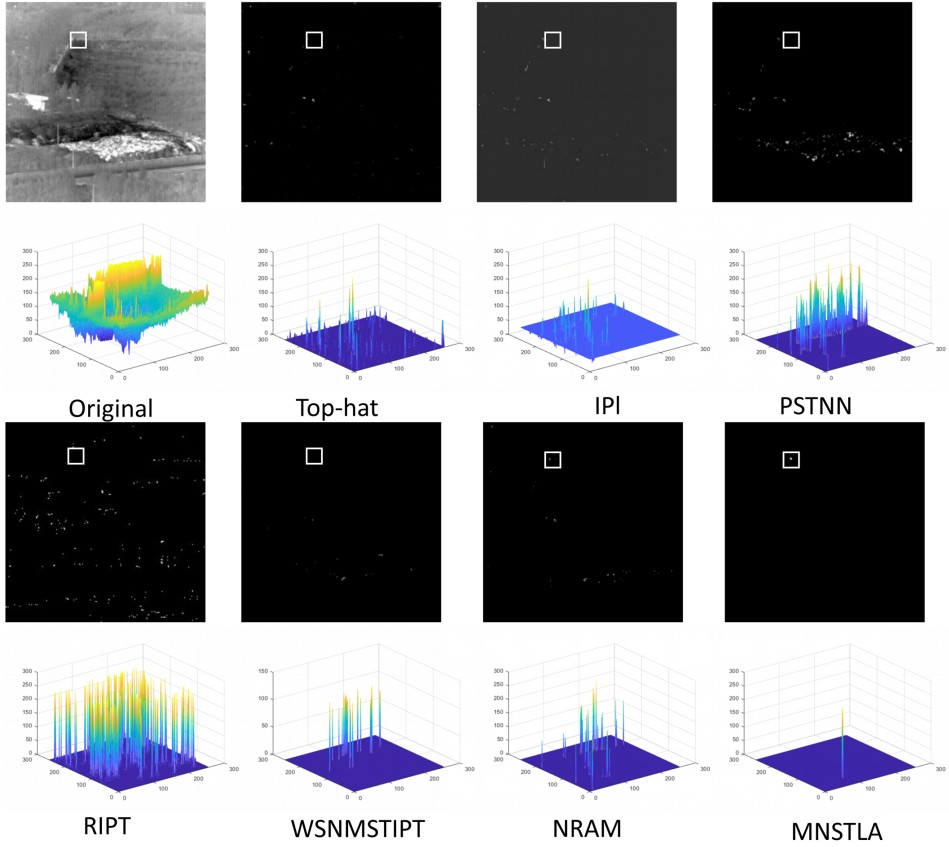

**Figure 9.** Infrared Sequence (e) Original image and Detection Results and the 3d grid diagrams.

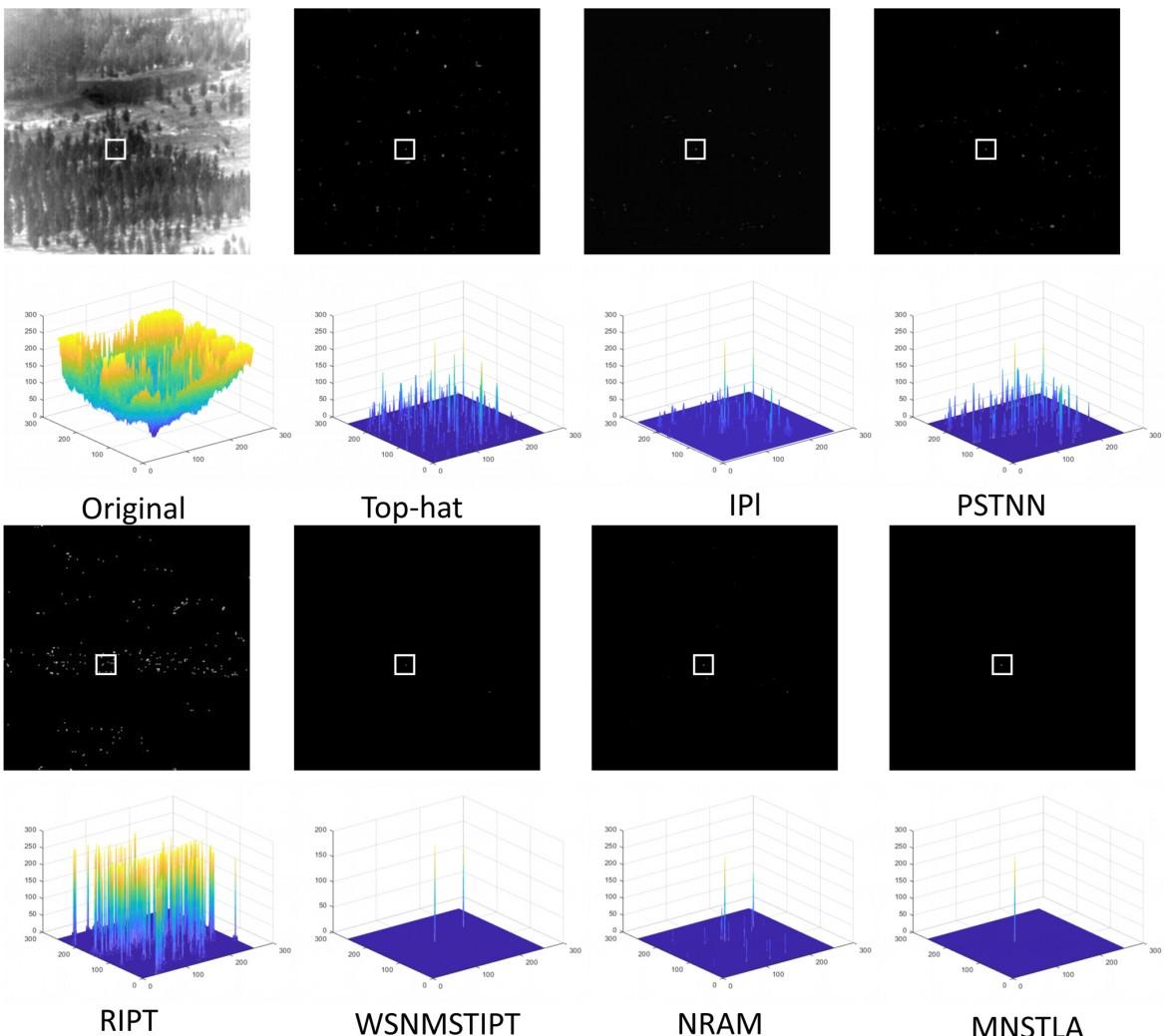

**Figure 10.** Infrared Sequence (f) Original image and Detection Results and the 3d grid diagrams.

It can be seen from Figures 5–10 that the RIPT model has the worst detection efficiency; the Top-Hat and PSTNN methods do enhance the targets, but edges and noise are introduced, which is mainly due to the assumption of fixed structural elements and a smooth background. Undoubtedly, among all the results from the test methods, the Top-Hat and PSTNN methods have the worst performance. This is because this contrast mechanism is not suitable for complex backgrounds. The IPI method is slightly better than the Top-Hat and PSTNN methods. The WSNMSTIPT models are on account of the IPI model and refer to the spatial-temporal information. Compared with the IPI model, although their false alarm rates are effectively reduced, not only do the images with dim targets selected from data sets 13 and 17 (corresponding to Figures 7 and 9) lose their targets, but also the images selected from the data sets with complex backgrounds lose their targets. Compared with the WSNMSTIPT models, the NRAM model does not consider the spatial-temporal information; it constructs the target-patches and background-patches according to the sparse feature of infrared target images. It can be seen from Figures 5–10 that, compared with the IPI model, the NRAM method not only effectively reduces the false alarm rate but also effectively enhances the strong edges. Therefore, the potential target points are also enhanced, and a better detection rate is achieved compared with the IPI model. The MNSTLA model proposed herein constructs, on account of the NRAM model and the spatial-temporal information, a spatial-temporal tensor model of infrared dim moving targets that fully considers the correlation between the frames of infrared dim moving

targets and can further reduce the false alarm rate and improve the detection efficiency of infrared dim moving targets.

### 4.4. Objective Evaluation for Different Scenes

We evaluate the performance of the MNSTLA model using the LCG and the BSF. The experimental results of the six actual sequences (Figures 5 and 6) are shown in Table 3. It can be seen that the method presented here can achieve the best values.

**Table 3.** Average Values of BSF and LCG of the Six Infrared Sequence Images Obtained by the Methods.

| Methods | a | | b | | c | | d | | e | | f | |
|---|---|---|---|---|---|---|---|---|---|---|---|---|
| | BSF | LCG | BSF | LCG | BSF | LCG | BSF | LCG | BSF | LCG | BSF | LCG |
| Top-Hat | 7.73 | 5.94 | 3.28 | 6.76 | 7.86 | 1.67 | 9.66 | 7.53 | 10.25 | 3.64 | 7.34 | 3.45 |
| PSTNN | 3.85 | 1.23 | 3.86 | 8.20 | 4.16 | 1.18 | 3.67 | 2.43 | 4.14 | 3.16 | 3.14 | 2.99 |
| IPI | 3.35 | 1.70 | 2.30 | 5.65 | 3.45 | 1.06 | 3.19 | 3.18 | 5.61 | 2.37 | 2.02 | 1.94 |
| RIPT | 0.92 | 3.11 | 0.72 | 3.16 | 1.76 | 1.29 | 1.62 | 2.01 | 1.26 | 1.29 | 0.56 | 1.93 |
| WSNMSTIPT | 5.16 | 6.22 | 2.08 | 22.35 | 4.26 | 2.36 | 5.08 | 2.86 | 3.46 | 4.16 | 3.29 | 3.38 |
| NRAM | 26.45 | 1.235 | 23.74 | 6.39 | 7.08 | 1.68 | 18.16 | 16.18 | 9.31 | 2.17 | 10.67 | 4.86 |
| MNSTLA | **61.25** | **8.353** | **36.29** | **26.58** | **63.42** | **6.98** | **39.61** | **7.69** | **54.36** | **5.93** | **53.17** | **5.29** |

Table 3 shows the average BSF and LCG of different methods on the six infrared image sequences. The Top-Hat and PSTNN methods have the lowest BSF and LCG values, and the corresponding background suppression capability is the worst. The IPI, RIPT, and WSNMSTIPT models have achieved good results in the six infrared image sequences, among which the RIPT and WSNMSTIPT models are slightly better than the IPI models in terms of performance; the NRAM model obtained a higher BSF value in the first sequence, but compared with the RIPT and WSNMSTIPT model, its background suppression ability is still not ideal; the MNSTLA model proposed herein achieved the highest BSF value on all six infrared image sequences, which means the robustness and efficiency of background suppression are better. In terms of LCG, this method has the highest LCG value and the best target enhancement of the six image sequences. From the evaluation results, it can be seen that the LCG and BSF values of the MNSTLA model proposed herein are much higher than those of other methods, indicating that it has great advantages in object enhancement and that the signal-to-noise ratio of images is improved effectively.

In order to compare the above optimization methods more objectively, the comparison of the ROC curves of the sequences 1–6 is shown in Figure 11. It is found in the study that the RIPT was the worst performer and that the Top-Hat method and the PSTNN method are not satisfactory. The IPI model achieved good results on the six infrared image sequences, and the WSNMSTIPT methods are slightly better than the IPI model in terms of performance. The detection rate of the NRAM model is not as high as that of the WSNMSTIPT models, and this is because the NRAM model does not consider the temporal-spatial information. Finally, under the same false alarm ratio, the MNSTLA model proposed herein achieved the highest detection probability, which means that the proposed MNSTLA model has better performance than that of any of the other models.

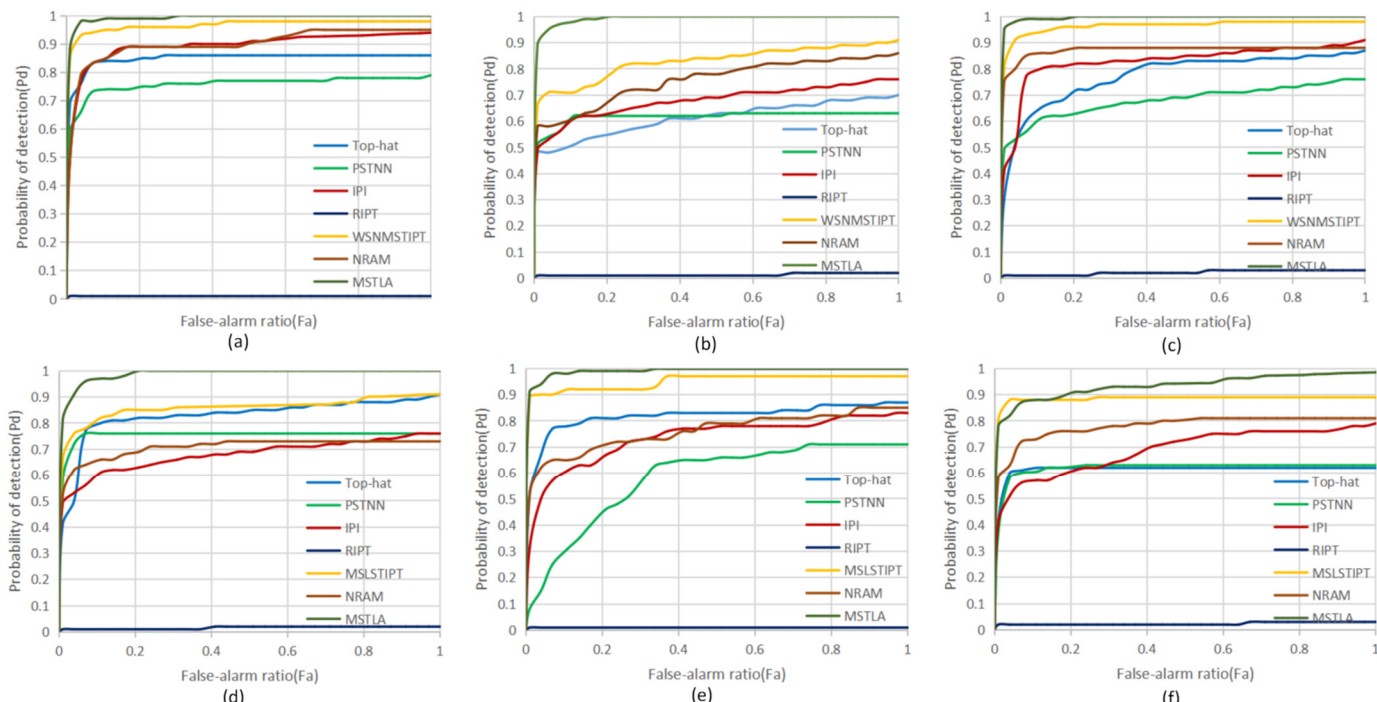

**Figure 11.** This is a figure. Schemes follow the same formatting. ROC curves of Six Image Sequences (**a–f**) Detected by Different Methods.

## 5. Discussion

The non-local auto-correlation on account of the infrared background and the target's sparsity has been extensively employed in the field of infrared tiny target detection. When the infrared image is homogeneous, a classical IPI effectively represents low-rank patch-background matrices using the nuclear norm. Larger solitary values really hold more information and visual detail. In other words, the complex infrared image is too complicated for the nuclear standard to handle, resulting in residual error and a blurry backdrop after reconstruction because of the rich details.

Currently, the majority of approaches concentrate on the priori backdrop and target, but this does not effectively separate the target from the background. In order to address the residual performance issue, RIPT proposes the structure tensor. The case of a poor signal-to-noise ratio, which leads to a lack of structure information and then target loss, is ignored by RIPT in complicated scenes. The NRAM model, on account of the IPI model, introduces a tighter rank proxy.

Based on the NRAM model, this article initially constrains the low-rank background using the tensor kernel norm rather than the rank function. The proposed MNSTLA model and other cutting-edge techniques can effectively suppress the interference caused by dynamic background and object moving on the foreground extraction and also show good performance in background suppression and object enhancement, according to qualitative and quantitative comparisons.

## 6. Conclusions

The robustness and effectiveness of a detection method for infrared point and moving targets are of great importance to the requirements of the early warning system. However, it is difficult to detect infrared dim and point targets, especially the point and moving targets. Therefore, we proposed a detection method using the minimization of a non-convex spatial-temporal tensor low-rank approximation for infrared points and moving targets. Our method introduces the concept of a spatial-temporal tensor on the basis of the non-convex rank approximation method. The experimental results on the real sequence

data sets in different scenes illustrate that this method is robust and effective in detecting infrared points and moving targets, and is less affected by background changes and poor image quality.

By the above discussion, while the MNSTLA model has a lower false alarm rate, the comparison is based on single target detection. However, in the IRST system, for multi-target detection of infrared sequence images or infrared videos, the spatial and temporal information is extremely crucial to improve the detection rate of dim and point targets and reduce the false alarm rate. Therefore, constructing a model that can simultaneously use the spatial-temporal information of infrared image sequences for multi-target detection is the focus of our further research. Therefore, we will consider combining the spatial-temporal information with the existing method in the follow-up research in the hopes of realizing the multi-target detection, improving the efficiency of target detection, and reducing the false alarm rate.

**Author Contributions:** Data curation, K.W.; Investigation, K.W.; Methodology, K.W.; Supervision, D.J. and X.L.; Validation, K.W.; Writing—review & editing, K.W. and D.J. and L.Y. All authors have read and agreed to the published version of the manuscript.

**Funding:** This research was funded by Youth Project of Applied Basic Research Project, grant number 2013FD016.

**Institutional Review Board Statement:** Not applicable.

**Informed Consent Statement:** Not applicable.

**Data Availability Statement:** No new data were created or analyzed in this study. Data sharing is not applicable to this article.

**Conflicts of Interest:** The authors declare no conflict of interest.

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
