# Peer review of "Infrared Small and Moving Target Detection on Account of the Minimization of Non-Convex Spatial-Temporal Tensor Low-Rank Approximation under the Complex Background"

_applsci, doi:10.3390/app13021196_

Round 1

Reviewer 1 Report

1. The research method is not clearly stated in the abstract, so it is necessary to state it precisely.
2. I suggest that the sections like section 2.1 (Spatial-Temporal Patch Tensor Model) which state the definitions, should be stated in a new section after the introduction and before the related works under the title of Theoretical Framework.
3. In the related works section, researches related to the subject of the article, their advantages, limitations and disadvantages should be addressed. Also, briefly explain how this research solves the disadvantages and limitations of previous methods.

Reviewer 2 Report

Title is too long. It should be revised.

Abstract should be revised and significance along with contributions should be added in the abstract.

Equations 1 and 2 are repeated. 

In related work, authors would benefit with a critical evaluation table instead of simple discussion of papers separately.

Figure 1 showing flowchart is not clear. it should be redrawn for better visibility. 

In results section, comparison with state of the art is completely missing.

Authors need to incorporate their contributions in the Introduction section.

In section 5, it should be Conclusion and Future work and add a paragraph for future directions in it.

Reviewer 3 Report

1)      The importance of the design carried out in this manuscript can be explained better than other important studies published in this field. I recommend the authors to review other recently developed works.

2)      "Discussion" section should be edited in a more highlighting, argumentative way. The author should analysis the reason why the tested results is achieved.

3)      It will be helpful to the readers if some discussions about insight of the main results are added as Remarks.

4)  "Discussion" section should be edited in a more highlighting, argumentative way. The author should analysis the reason why the tested results is achieved.

5)      Similarly, "Conclusion" section should be rearranged. Taking advantage of these results, striking suggestions can be made for future studies.

6)     The authors should clearly emphasize the contribution of the study. Please note that the up-to-date of references will contribute to the up-to-date of your manuscript. The studies named- Faults Detection With Image Processing Methods In Textile Sector, Soft Tissue Sacromas Segmentation using Optimized Otsu Thresholding Algorithms, ANALYSIS OF BURIED OBJECTS WITH DEEP LEARNING METHOD IN GPR IMAGING SYSTEMS in the study or to indicate the contribution in the "Introduction" section.

Round 2

Reviewer 2 Report

Authors have addressed most of my comments. 

Reviewer 3 Report

All corrections were well-done